# The Histamine H_4_ Receptor Participates in the Neuropathic Pain-Relieving Activity of the Histamine H_3_ Receptor Antagonist GSK189254

**DOI:** 10.3390/ijms232214314

**Published:** 2022-11-18

**Authors:** Vittoria Borgonetti, Nicoletta Galeotti

**Affiliations:** Department of Neurosciences, Psychology, Drug Research and Child Health (NEUROFARBA), Section of Pharmacology and Toxicology, University of Florence, Viale G. Pieraccini 6, 50139 Florence, Italy

**Keywords:** neuropathic pain, histamine H3 receptor antagonist, histamine H4 receptors, spinal cord, locus coeruleus, spared nerve injury

## Abstract

Growing evidence points to the histamine system as a promising target for the management of neuropathic pain. Preclinical studies reported the efficacy of H_3_R antagonists in reducing pain hypersensitivity in models of neuropathic pain through an increase of histamine release within the CNS. Recently, a promising efficacy of H_4_R agonists as anti-neuropathic agents has been postulated. Since H_3_R and H_4_R are both localized in neuronal areas devoted to pain processing, the aim of the study is to investigate the role of H_4_R in the mechanism of anti-hyperalgesic action of the H_3_R antagonist GSK189254 in the spared nerve injury (SNI) model in mice. Oral (6 mg/kg), intrathecal (6 µg/mouse), or intra locus coeruleus (LC) (10 µg/µL) administration of GSK189254 reversed mechanical and thermal allodynia in the ipsilateral side of SNI mice. This effect was completely prevented by pretreatment with the H_4_R antagonist JNJ 10191584 (6 µg/mouse i.t.; (10 µg/µL intraLC). Furthermore, GSK189254 was devoid of any anti-hyperalgesic effect in H_4_R deficient mice, compared with wild type mice. Conversely, pretreatment with JNJ 10191584 was not able to prevent the hypophagic activity of GSK189254. In conclusion, we demonstrated the selective contribution of H_4_R to the H_3_R antagonist-induced attenuation of hypernociceptive behavior in SNI mice. These results might help identify innovative therapeutic interventions for neuropathic pain.

## 1. Introduction

Neuropathic pain is a disabling condition with a prevalence rate in the general population that lies between 7% and 10% [1]. Despite advances in the understanding of the underlying causes and mechanisms leading to the development and maintenance of neuropathic pain, current therapies have substantial limitations in terms of efficacy and safety profile and about 40% of patients who suffer from chronic pain did not achieve adequate pain control [2]. Consequently, there is an urgent need for the identification of new therapeutic interventions to improve management of neuropathic pain to improve the outcome for pain suffering patients.

Growing evidence indicates the histamine system as a therapeutic target for the management of neuropathic pain and recent studies have supported the use of selective ligands of histamine H_3_ receptor (H_3_R) and H_4_ receptor (H_4_R) for the treatment of neuropathic pain [3,4,5]. H_3_R is primarily expressed in the central nervous system (CNS) with a prominent localization in nociceptive and pain-control pathways, such as the thalamus, prefrontal cortex, periaqueductal gray area, spinal cord, and dorsal root ganglia, which suggests its functional involvement in the regulation of nociceptive transmission [6]. H_3_R has a prevalent presynaptic expression and is recognized as a presynaptic autoreceptor involved in the negative regulation of the release of histamine [5] and a heteroreceptor, regulating the release of neurotransmitters such as acetylcholine [7], dopamine [8], noradrenaline [9], and serotonin [10]. Preclinical studies showed the efficacy of H_3_R antagonists in reducing mechanical and thermal pain hypersensitivity in different model of neuropathic pain (i.e., chronic constriction injury, spinal nerve ligation) [3,11,12,13,14]. Increasing evidence also indicates H_4_R as an interesting target for the treatment of neuropathic pain. Even though its principal role appears to be the regulation of immune/inflammatory responses [15], a neuronal expression of this receptor has also been postulated. Recent reports indicate the presence of H_4_R on peripheral sensory nerves, in the DRG, and in the spinal cord, especially laminae I and II [16,17], supporting the involvement of H_4_R in the modulation of pain processing [16,18]. H_4_R pharmacological modulators have documented the involvement of H_4_R in acute pain [19] and persistent inflammatory pain [11]. More recently, H_4_R agonists have been shown to counteract trauma-induced neuropathic pain, reducing both oxidative stress and pro-neuroinflammatory pathways [16,20] and studies on H_4_RKO mice have further supported a role for H_4_R in neuropathic pain conditions [18].

H_3_R and H_4_R have both a neuronal expression and a higher affinity (nM range) for histamine than H_1_R and H_2_R (μM range) [21]. Although the localization of H_4_R on neuronal cells has not been fully elucidated, expression of H_3_R and H_4_R on the opposite sides of the synaptic cleft may contribute to their effects in neuropathic pain perception. Thus, the aim of the present study is to investigate the role of H_4_R in the mechanism of anti-hyperalgesic action of H_3_R antagonists in a model of peripheral neuropathy, the spared nerve injury (SNI) model in mice. We focused our investigation on the H_3_R antagonist GSK189254, compound with high selectivity and high affinity for human (pKi = 9.59 − 9.90) and rat (pKi = 8.51 − 9.17) H_3_R [22]. In addition, GSK189254 is a compound of therapeutic relevance being endowed with analgesic properties in neuropathic pain states in both preclinical and clinical studies [5,23] that progressed into phase I clinical trial (clinicaltrials.gov).

## 2. Results

### 2.1. GSK189254 Attenuates Pain Hypersensitivity in SNI Mice through H_4_R Activation

The murine model of neuropathic pain induced by SNI was reproduced to investigate the involvement of H_4_R in the anti-hyperalgesic activity of the H_3_R antagonist GSK189254. Time-course studies showed that the SNI procedure induced an intense and long-lasting thermal (Figure 1A) and mechanical (Figure 1B) allodynia-like behaviors measurable in the non-injured sural nerve skin territory of the ipsilateral side. Starting from day 3 post-surgery, the paw withdrawal threshold to non-noxious mechanical thermal or mechanical stimulus was significantly decreased with respect to the same measure performed before the nerve damage (baseline values) or to the uninjured contralateral side at every time point. Similarly, no decrease in the pain threshold was detected in the sham-operated ipsilateral side, used as SNI control group.

Oral treatment with the H_3_R antagonist GSK189254 6 mg/kg reversed mechanical (Figure 2A) and thermal (Figure 2B) allodynia. The dose of 3 mg/kg was ineffective. The anti-hyperalgesic activity of GSK189254 was completely abolished by the H_4_R antagonist JNJ 10191584 (6 mg/kg p.o.). The dose of the H_4_R antagonist, when administered alone, was devoid of any effect on mouse pain threshold.

To further elucidate the role of H_4_R in the anti-hyperalgesic effect induced by the investigated H_3_R antagonist, H_4_RKO mice were used. We first detected the baseline thermal and mechanical sensitivity. Both H_4_R-deficient and wild-type (Wt) mice developed thermal and mechanical hypersensitivity in the ipsilateral paws following SNI (Figure 2C,D). Oral administration of GSK189254 (6 mg/kg) showed anti-hyperalgesic activity in Wt mice whereas in H_4_RKO mice no significant modification of the mechanical (Figure 2C) and thermal allodynia (Figure 2D) in the ipsilateral side was observed. Double staining immunofluorescence micrographs from spinal cord samples of Wt mice showed the colocalization of H_4_R with NeuN, a neuronal marker, confirming the H_4_R neuronal localization. Double-labeling immunostaining of H_4_R with DAPI (4′,6-diamidino-2-phenylindole), a well-known nuclei marker, to assess the subcellular localization of H_4_R showed its expression in the plasma membrane and its absence in the nucleus. Slices from H_4_RKO mice did not show any H_4_R immunostaining (Figure 2E).

To evaluate whether this activity was related to the modulation of the histaminergic system within the central nervous system and to exclude the involvement of peripheral mechanisms, GSK189254 was administered directly to the central nervous system (CNS). Intrathecal (i.t) administration of GSK189254 (6 µg/mouse) ameliorated mechanical (Figure 3A) and thermal (Figure 3B) allodynia, and these effects were prevented by treatment with the H_4_R antagonist JNJ 10191584 (6 µg/mouse). Intra locus coeruleus (LC) administration of GSK189254 (10 µg/µL) produced a similar efficacy in attenuating mechanical (Figure 3C) and thermal (Figure 3D) allodynia. Consistently, intra-LC administration of JNJ 10191584 (10 µg/µL) abolished the anti-hyperalgesic activity of the H_3_R antagonist.

H_3_R antagonists increase histamine release within the CNS by blocking H_3_ autoreceptors [24], thus producing analgesia. To further verify the specific involvement of H_4_R in the modulation of neuropathic pain by GSK189254, we tested the effects produced by centrally delivered histamine and VUF 8430, a H_4_R selective agonist. I.t. administration of histamine (40 µg/mouse) reversed both mechanical (Figure 4A) and thermal (Figure 4B) allodynia in SNI mice. Comparable effects were produced by the i.t administration of VUF 8430 (20 µg/mouse) (Figure 4C,D). Pretreatment with the H_4_R antagonist JNJ 10191584 (6 µg/mouse i.t.) completely prevented histamine- and VUF 8430-induced antiallodynic activity (Figure 4A–D) with no effect on the uninjured contralateral side.

### 2.2. Lack of Contribution of H_4_R to the Anorexiant Effect of GSK 189254

Consistent with literature data on H_3_R antagonists [25], we observed a hypophagic effect for GSK189254 (6 mg/kg p.o). The cumulative food-consumption was measured in 12 h-fasted mice who showed a constant increase of food eaten over a 60 min period of observation. Treatment with GSK189254 significantly reduced the food consumption and this anorexiant effect was not reversed by pretreatment with the H_4_R antagonist JNJ 10191584 (6 mg/kg p.o) (Figure 5A). The lack of a prominent role of H_4_R in the hypophagic activity of the H_3_R antagonist was further confirmed in H_4_RKO mice that showed results comparable to Wt mice (Figure 5B).

### 2.3. Lack of Locomotor Side Effects

GSK189254, at the highest effective dose (6 mg/kg p.o.), did not induce any visible sign of altered gross behavior or poor health. Evaluation of locomotor behavior by specific tests revealed the absence of any significant motor impairment. Motor coordination was evaluated by use of the rotarod test. Treated mice did not show any impairment in the motor coordination and the number of falls from the rotating rod was identical to that of untreated control mice (Figure 6A). Spontaneous mobility and exploratory activity was evaluated by use of the hole board test showing the absence of any significant increase of both parameters in comparison with the control group (Figure 6B,C). Thus, attenuation of the nociceptive behavior in SNI mice promoted by GSK189254 was not compromised by the induction of undesirable side effects.

## 3. Discussion

In the present study, we investigated the role of H_4_R in the mechanism of anti-neuropathic action of the H_3_R antagonist GSK189254. The results obtained illustrated that the stimulation of H_4_R is involved in the relief from pain hypersensitivity in a mouse model of peripheral mononeuropathy, the SNI model.

GSK189254 has been reported to reduce mechanical and cold hypersensitivity associated with neuropathic pain in the chronic constriction injury model or spinal nerve ligation model in rats after a single intraperitoneal dose [11,26]. In addition, repeated, orally delivered doses of GSK189254 significantly reduced rat paw withdrawal threshold to mechanical stimuli in trauma-induced and virally induced neuropathic pain [26]. Consistent with these previous results, we showed the efficacy of a single oral administration of GSK189254 in attenuating mechanical and thermal allodynia in the SNI model in mice. No effect was detected following GSK189254 treatment in the uninjured contralateral side, indicating a prominent contribution of H_3_R-mediated events in chronic pain states rather than in the physiological perception of nociception. This hypothesis is further corroborated by results from behavioral studies that indicated the lack of induction of analgesia in healthy mice by H_3_R antagonists [3,27] and an unaltered mechanical pain threshold in H_3_RKO mice [28].

H_3_R is predominantly expressed in neurons with a presynaptic localization [24], acting as autoreceptor [24,29]. Since an increase of histamine levels in the CNS has been reported to produce analgesia in neuropathic pain states [5], the increase of pain threshold induced by the blockade of the H_3_R can be related to the regulation of histamine levels in the CNS. Recently, the involvement of the H_4_R activation in the analgesic effect of histamine has been reported [20,30]. In line with this observation, pretreatment with the potent and selective H_4_R antagonist JNJ 10191584 [31] prevented the GSK189254 anti-neuropathic effect against both mechanical and thermal hypersensitivity. Experiments conducted on H_4_RKO mice showed the lack of efficacy of GSK189254 in comparison with Wt mice.

Localization of H_3_R and H_4_R on neuronal structures involved in the modulation of pain perception further supports their role in nociceptive transmission. Immunohistochemical studies identified H_3_R on medium-size cell bodies in dorsal root ganglia (DRG) and in dorsal horn laminae I, II, and V [6]. Interestingly, a similar localization was reported for H_4_R. Recent studies indicate the presence of H_4_R in the small- and medium-diameter cells of DRG, and in the laminae I and II of the spinal cord [16,17]. In addition, H_3_R and H_4_R represent the histamine receptors with the highest affinity for histamine (nM range vs. μM range showed by H_1_R and H_2_R) [21]. Interestingly, results from H_4_RKO mice showed an unaltered physiological pain threshold, with a pain sensitivity comparable with that of Wt mice. Conversely, an enhanced hypersensitivity to mechanical and thermal stimuli compared to the allodynia of Wt control mice was detected in H_4_RKO following the SNI procedure [18]. Thus, both the H_4_R expression in neuronal structure involved in the regulation of neuropathic pain and the pain-relieving effects induced by H_4_R stimulation suggest that H_4_R activation is specifically involved in the regulation of pain hypersensitivity associated with pathological neuropathic chronic pain conditions, with a similar pharmacological profile to a H_3_R antagonism.

The spinal cord has been observed to be an important site of action of GSK189254. Indeed, both systemic and i.t. administration of the compound attenuated pain perception in models of neuropathic pain and osteoarthritis [13,32]. In addition, another important site of action of GSK189254 has been reported to be the LC [27]. Consistent with these observations, i.t and intra-LC administration of GSK189254 reduced both mechanical and thermal allodynia in SNI mice. The nociceptive behavior attenuation was antagonized by the H_4_R antagonist JNJ 10191584 administered i.t. or intra-LC, respectively, further confirming the involvement of H_4_R-mediated mechanisms in the antinociceptive effect induced by the H_3_R blockade within the SNC.

Concerning the mechanism of analgesic action of H_3_R antagonists, it has been proposed that a critical role is played by the noradrenergic system, mediated, at least in part, by the modulation of α2 adrenoceptors in the spinal cord and LC. Numerous studies indicate that descending inhibitory pathways diminish nociceptive transmission in the dorsal horn of the spinal cord through the involvement of noradrenaline release in the spinal cord from the LC [33,34]. Thus, in the presence of neuropathic pain, the activation of the LC descending pathway might promote the control of pain hypersensitivity by activating spinal α2-adrenoreceptors. The attenuation of nociceptive behavior by both the systemic and i.t. administration of GSK189254 was reversed by pretreatment with the α1/2 adrenoceptor antagonist phentolamine [32] and the facilitation of endogenous release of histamine in the LC by H_3_R blockade has been reported to lead to neuropathic hypersensitivity inhibition through the regulation of descending noradrenergic pathways [35]. Recently, it has been reported that i.t. administration of the α2-adrenoceptor agonist clonidine, a drug used in the management of acute and chronic pain [36,37,38] and approved by the United States Food and Drug Administration (FDA) for epidural administration in treating neuropathic cancer pain and attenuated mechanical and thermal allodynia in SNI mice. This anti-neuropathic effect was prevented by the H_4_R antagonist JNJ 10191584 administered intra LC and in H4R deficient mice clonidine failed to ameliorate the nociceptive behavior [30]. In addition, H_4_R immunostaining was detected on noradrenergic neurons expressing phosphorylated cAMP response element binding protein (CREB), a marker of neuronal activation [30]. Present finding let hypothesize that application of GSK189254 increases histamine release, which activates H_4_R and subsequently causes coeruleospinal noradrenergic neuron activity resulting in suppressing pain. These data further support a correlation between H_3_R and H_4_R and help better define the role of the histaminergic system within the LC-spinal tract in the modulation of pain hypersensitivity in neuropathic pain states.

Antagonists of sigma 1 receptor are effective in models of neuropathic pain [39,40]. Several H_3_R antagonists have been shown to occupy the sigma 1 receptor at anti-hyperalgesic doses, letting hypothesize that the in vivo activity of H_3_R antagonists may be due to a combination of both mechanisms. However, GSK189254 displayed high H_3_R selectivity [41] suggesting that the H_3_R antagonism could prominently contribute to the in vivo efficacy.

Brain histamine is largely involved in eating behavior. A histamine-induced loss of appetite has long been described and several preclinical studies showed that H_3_R antagonists reduced the food intake, body weight, and blood glucose level in obese animals [25]. In addition, hypophagic activity has been described following H_4_R activation [18,19]. GSK189254, administered at analgesic doses, reduced food consumption in mice fasted for 12 h. The investigation into a potential role of H_4_R in the mechanism of hypophagic activity of the H_3_R antagonist showed the lack of prevention of the anorexiant effects by pretreatment with an H_4_R antagonist. Furthermore, the reduction of food consumption by GSK189254 was not abolished in H_4_RKO mice, confirming the lack of a prominent role of H_4_R in the mechanism of anorexiant activity of GSK189254.

In conclusion, our results demonstrate the contribution of H_4_R to the attenuation of the hypernociceptive behavior in SNI mice, a model of trauma-induced neuropathic pain, induced by an H_3_R antagonism. The pain-relieving activity of the H_3_R antagonist GSK189254 is prevented by the oral, i.t., or intra-LC administration of the H_4_R antagonist JNJ 10191584 and abolished in H_4_RKO mice. These data better elucidate the cellular and molecular mechanisms underlying the pain modulating activity of H_3_R antagonists in neuropathic pain states and further define the role of neuronal H_4_R in nociception. These results might help better identify innovative therapeutic interventions for neuropathic pain conditions.

## 4. Materials and Methods

### 4.1. Drugs

VUF 8430 dihydrobromide and histamine dihydrochloride (Tocris Bioscience, Bristol, UK) were dissolved in saline (0.9% NaCl) immediately before use; GSK189254 (6-((3-cyclobutyl-2,3,4,5-tetrahydro-1H-3-benzazepin-7-yl)oxy)-Nmethyl-3-pyridinecarboxamide) (MedChemExpress, Monmouth Junction NJ, USA) and JNJ 10191584 maleate (Tocris Bioscience, Bristol, UK) were dissolved in 5% DMSO.

### 4.2. Experimental Design 

All analgesic drugs were administered 30 min before testing. For oral administrations, drugs were dispersed in 1% sodium carboxymethylcellulose and administered by gavage 30 min before behavioral testing (Figure 7). Doses and administration schedule were chosen on the basis of time-course and dose-response curves performed in our laboratory [19,20].

### 4.3. Animals

Experiments were performed on male BALB/C wild type (WT) and histamine H_4_R knockout (H4^−/−^) (16–18 g) mice. H4^−/−^ mice were generated by Lexicon Genetics (Woodlands Park, TX, USA) and provided by Janssen Research & Development (LLC, La Jolla, CA, USA) and back crossed to BALB/c background. Corresponding WT mice were obtained from Envigo (Udine, Italy). Animals were housed under standard conditions [20].

All studies involving animals are reported in accordance with the ARRIVE guidelines for experiments involving animals [42]. Protocols were designed to minimize the number of animals used and their suffering. The number of animals per experiment was based on a power analysis [43] and calculated by G power software. Eight animals per group were used.

### 4.4. Microinjection into the Locus Coeruleus

Animals receiving intra locus coeruleus (LC) injections received stereotaxic placement of a stainless-steel cannula (26 gauge; PlasticsOne, Roanoke, VA, USA) under general anesthesia. Coordinate for LC placement were anteroposterior (AP), −5.45 mm from bregma; mediolateral (ML), 1.28 mm; and dorsoventral (DV) −3.65 mm from the surface of the skull [30]. The volume of intra LC injections was 0.1 μL. To visually verify the placement of the cannula, at the end of the experiment, mice received an intra-LC injection of methylene blue (0.1 μL), were euthanized, and the brain removed and sectioned.

### 4.5. Intrathecal Injection

Intrathecal (i.t.) administration was performed as previously described [44]. Mice received a single i.t. injection and drug concentrations were prepared in such a way that the necessary dose could be administered in a volume of 5 µL per mouse.

### 4.6. Spared Nerve Injury (SNI)

Spared nerve injury (SNI) mono-neuropathy was applied in mice [44]. Briefly, in anaesthetized mice (4% isoflurane in O_2_/N_2_O (30:70 *v*/*v*)), a small incision was made on the lateral surface of the thigh and the sciatic nerve was exposed; the three peripheral branches (sural, common peroneal, and tibial nerves) of the sciatic nerve were exposed without stretching nerve structures and separated. Both the common peroneal and tibial nerves were tightly ligated and cut, leaving the sural nerve intact. In the sham group, the same procedure was applied only without the cut of sciatic nerve branches.

### 4.7. Nociceptive Behaviour

Animals were habituated to the testing environment daily for at least 2 days before baseline testing. Behavioral testing was performed before surgery to establish a baseline for comparison with postsurgical values. For time-course studies, nociceptive responses to mechanical and thermal stimuli were measured daily from day 3 to day 28 after nerve surgery. Nociceptive recordings to evaluate the effects of drug treatments were conducted on day 14 post-surgery (Figure 7). Each mouse served as its own control, the responses being measured both before and after surgery. All testing was performed with a blind procedure.

#### 4.7.1. Von Frey Test

The Dynamic Plantar Aesthesiometer (Ugo Basile) was used to measure mechanical allodynia [20]. After 1 h acclimatization, an automated testing device delivered a mechanical stimulus to the plantar surface of the hind paw of the animal. The paw withdrawal stopped the mechanical stimulus, and the force was recorded to the nearest 0.1 g and used as a parameter to define the mouse mechanical pain threshold. The responses were measured both before and after administrations. Both ipsilateral (injured) and contralateral (uninjured) paws were tested.

#### 4.7.2. Hargreaves’ Plantar Test

The Hargreaves’ device (plantar test apparatus; Ugo Basile, Comerio, Italy) was used to measure the thermal nociceptive threshold [18]. After 1 to 2 h acclimatization, the paw withdrawal latency in response to radiant heat (infrared) was measured. The paw withdrawal response to the radiant heat automatically stopped the stimulus, the time elapsed was measured in seconds and used as parameter to define the mouse thermal pain threshold. The responses were measured both before and after administrations. Both ipsilateral (injured) and contralateral (uninjured) paws were tested.

### 4.8. Locomotor Activity

#### 4.8.1. Rotarod Test

This test was used to evaluate motor coordination and balance. The apparatus is set to a rod-rotation speed of 16 r.p.m. The test begins when acceleration is started, and measurements are stopped every time the mouse fall off rod over a 30 s period. The motor coordination was defined as number of falls from the rod in 30 s [44].

#### 4.8.2. Hole-Board Test

The hole-board apparatus was used to measure the spontaneous locomotor activity (spontaneous mobility and exploratory activity). A single mouse is placed in the center of the board with regularly arranged holes on the floor for 5 min. The movement on the surface of the arena (spontaneous mobility) and the frequency of spontaneous hole-poking behavior (exploratory activity) by the mice was measured [44].

### 4.9. Evaluation of Food Consumption

Mice were tested after 12 h food deprivation while water remained available ad libitum. A weighed amount of standard chow pellets was placed in the food rack, and food consumption evaluated as the difference in weight between the amount of food initially provided and that left in the rack, including spillage in the cage. Food consumption was measured 15, 30, 45, and 60 min after the beginning of the test.

### 4.10. Immunofluorescence

Animals were anesthetized and perfused transcardially with 4% paraformaldehyde in 0.1 M phosphate buffer. After perfusion, spinal cord samples were processed for standard immunostaining as previously detailed [20].

### 4.11. Statistical Analysis

Results were expressed as mean ± s.e.m. One-way analysis of variance, followed by Tukey post hoc test and two-way analysis of variance followed by Bonferroni test were used for statistical analysis. GraphPad Software (LaJolla, CA, USA) was used to determine the statistical significance. *p* < 0.05 was statistically significant.

## Figures and Tables

**Figure 1 ijms-23-14314-f001:**
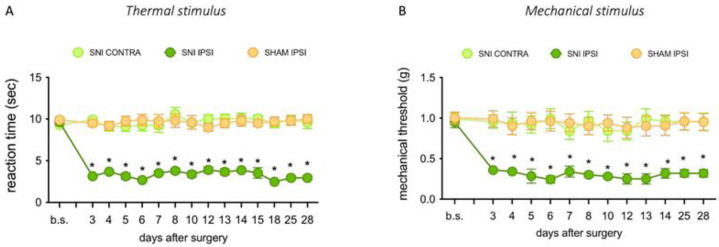
Hypernociceptive behavior of SNI mice. SNI produced a long-lasting thermal (**A**) and mechanical (**B**) allodynia from day 3 up to day 28 after surgery. Data are means ± SEM. * *p* < 0.001 versus contra (two-way ANOVA followed by Bonferroni test).

**Figure 2 ijms-23-14314-f002:**
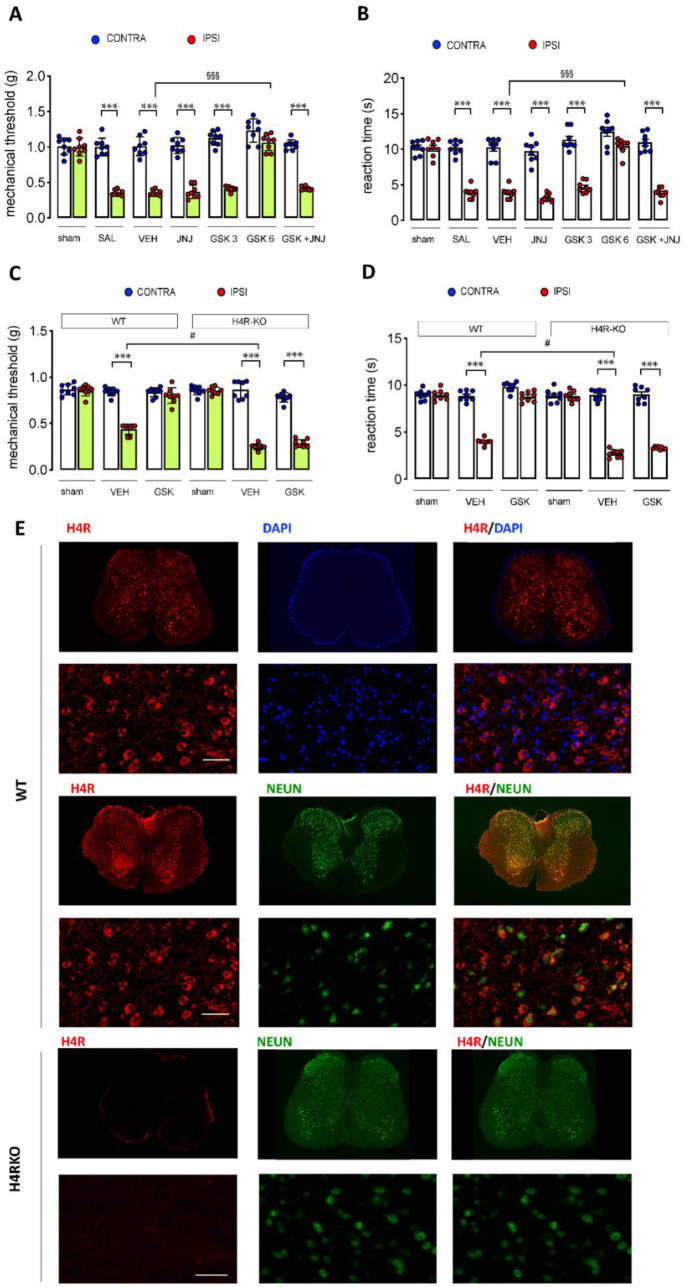
Attenuation of SNI-induced pain hypersensitivity by orally administered GSK189254 through a H_4_R-mediated mechanism. Oral administration of GSK189254 (GSK) (3 and 6 mg/kg p.o.) reversed thermal (**A**) and mechanical (**B**) allodynia in SNI mice. This effect was reversed by the H_4_R antagonist JNJ 10191584 (JNJ) (6 mg/kg p.o.). *** *p* < 0.001 versus contra, §§§ *p* < 0.001 versus ipsi (one-way ANOVA followed by Tukey test). Oral administration of GSK189254 (6 mg/kg p.o.) did not modify thermal (**C**) and mechanical (**D**) hypersensitivity in H_4_R deficient mice (H_4_RKO) while it showed anti-hyperalgesic activity in Wt mice. *** *p* < 0.001 versus Wt contra, # *p* < 0.05 versus Wt ipsi (one-way ANOVA followed by Tukey test). SAL: saline; VEH: vehicle (5% DMSO). (**E**) Double immunostaining photomicrographs (low magnification, top; high magnification, bottom) showed the expression in the plasma membrane and its absence in the nucleus. Nuclei were stained with DAPI. H_4_R immunolabelling illustrated the presence of H_4_R in the spinal cord of Wt mice that is co-expressed with NeuN, a neuronal marker. Slices from H_4_R-deficient mice (H_4_RKO) did not show any H_4_R immunostaining. Scale bar: 50 μm.

**Figure 3 ijms-23-14314-f003:**
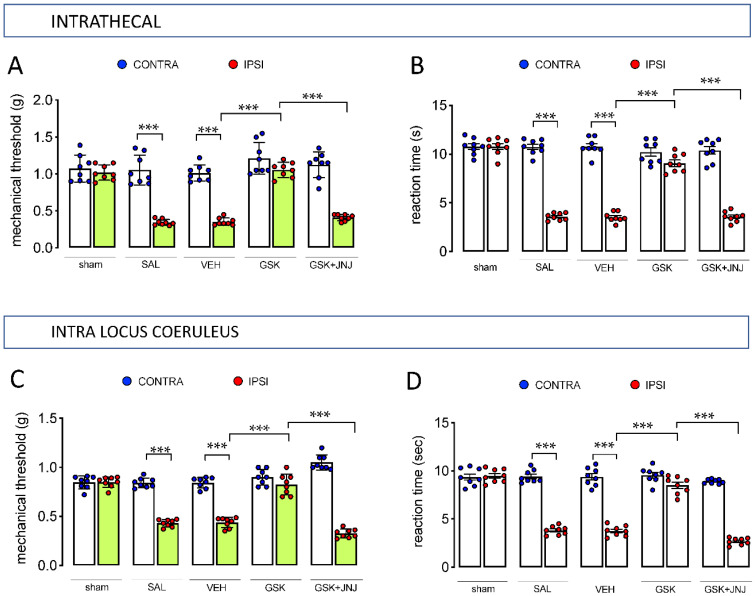
Anti-allodynic effect of intra CNS administration of GSK189254 in SNI mice mediated by H_4_R activation. Intrathecal (i.t.) administration of GSK189254 (GSK) (6 µg/mouse) reversed mechanical (**A**) and thermal (**B**) allodynia in SNI mice. This effect was antagonized by JNJ 10191584 (JNJ) (6 µg/mouse i.t.). Intra LC administration of GSK189254 (10 µg/µL) attenuated mechanical (**C**) and thermal (**D**). Intra-LC JNJ 10191584 (10 µg/µL) reversed this effect. SAL: saline; VEH: vehicle (5% DMSO). *** *p* < 0.001 (one-way ANOVA followed by Tukey test).

**Figure 4 ijms-23-14314-f004:**
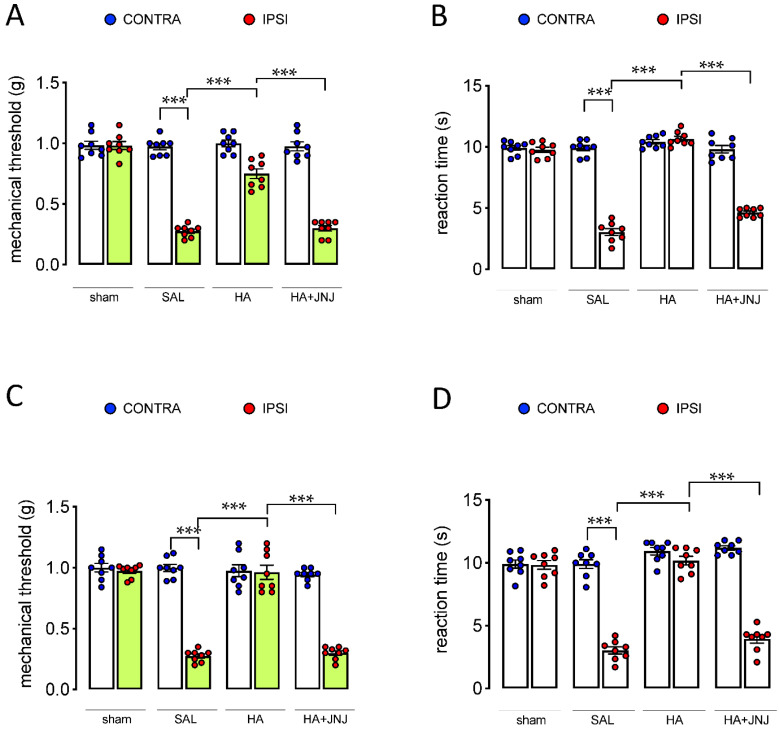
H_4_R-mediated anti-neuropathic effect of histamine (HA) and VUF8430 (VUF) on neuropathic pain symptoms in SNI mice. Intrathecal (i.t.) administration of HA (40 µg/mouse) reversed mechanical (**A**) and thermal (**B**) allodynia in SNI mice. This effect was antagonized by JNJ 10191584 (JNJ) (6 µg/mouse i.t.). I.t. injection of VUF8430 (20 µg/mouse) attenuated mechanical (**C**) and thermal (**D**) allodynia. JNJ 10191584 (6 µg/mouse i.t.) reversed this effect. SAL: saline. *** *p* < 0.001 (one-way ANOVA followed by Tukey test).

**Figure 5 ijms-23-14314-f005:**
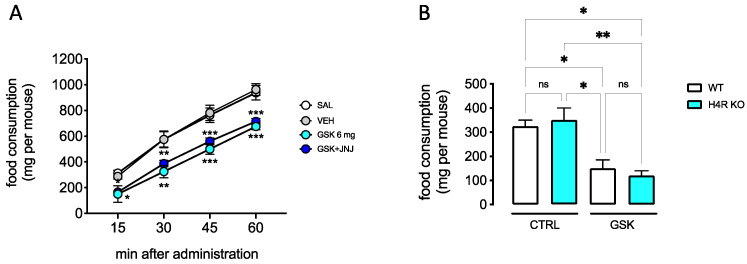
Hypophagic activity of GSK189254 through an H_4_R-independent mechanism. (**A**) Oral administration of GSK189254 (GSK) (6 mg/kg) reduced the cumulative amount of food consumption in 12 h-fasted mice. This effect was not antagonized by JNJ 10191584 (JNJ) (6 µg/kg p.o.) pretreatment. * *p* < 0.05, ** *p* < 0.01, *** *p* < 0.001 (two-way ANOVA followed by Bonferroni test). SAL: saline, VEH: vehicle (5% DMSO) (**B**) GSK189254 decreased the amount of food eaten in both Wt and H_4_RKO mice. Food consumption was measured 30 min after administration. CTRL: vehicle; ns: not significant, * *p* < 0.05, ** *p* < 0.01 (one-way ANOVA followed by Tukey test).

**Figure 6 ijms-23-14314-f006:**
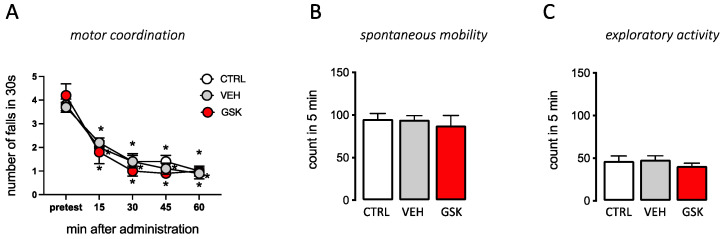
Lack of locomotor impairments by GSK189254. Oral administration of GSK189254 (GSK) (6 mg/kg p.o.) did not alter motor coordination (**A**), spontaneous mobility (**B**), or exploratory activity (**C**). CTRL: untreated mice; VEH: 5% DMSO. * *p* < 0.05 vs. pretest (two-way ANOVA followed by Bonferroni test).

**Figure 7 ijms-23-14314-f007:**
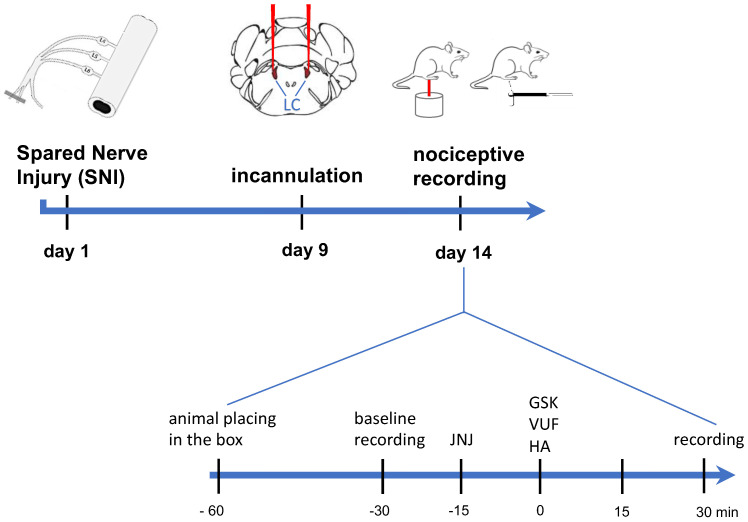
Administration and behavioral tests schedule.

## Data Availability

The data presented in this study are available on request from the corresponding author.

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
