# Peer review of "The Histamine H_4_ Receptor Participates in the Neuropathic Pain-Relieving Activity of the Histamine H_3_ Receptor Antagonist GSK189254"

_ijms, 2022, doi:10.3390/ijms232214314_

Round 1
Reviewer 1 Report
Dear Dr.,
Title: The histamine H4 receptor participates in the neuropathic pain-relieving activity of the histamine H3 receptor antagonist GSK189254
Manuscript ID: ijms-2014791
Overall comments: Authors described in this manuscript: the role of histamine receptor-4 (H4) in the neuropathic pain relief action with histamine receptor-3 (H3) antagonist (GSK189254) mice spared nerve injury (SNI) model. The overall manuscript is well written and it has novelty in this field of research. However, some of the contents need to improve for the better quality of this manuscript.
Specific comments:
1. The abstract is well written. The dose of histamine receptor antagonistic agents can be highlight in this section.
2. The introduction section has multiple numbers of small paragraphs. The logical and limited paragraph (by merging the small paragraph) is required for better understanding and to encourage the readers towards this research work.
3. The last statement “The results obtained showed that stimulation of H4R is selectively involved in the pain-relieving activity of GSK189254.” Need to place in the discussion section.
4. Result and discussion section: These sections are well written. The discussion section is too lengthy and can be concise the statement.
5. Conclusion section: This section is written well.
6. Materials and Methods: 4.1. Drugs section statement “All analgesic drugs were administered 30 min before testing. For oral administrations drugs were dispersed in 1% sodium carboxymethylcellulose and administered by gavage 30 minutes before behavioural testing (Figure 7). Doses and administration schedule were chosen on the basis of time-course and dose-response curves performed in our laboratory [21,23].” can place in a separate section as experimental design with figure 7.
7. Section 4.7.1. Rotarod test, line 402, some typo error is found, it needs to correct.
8. Section 4.7.2. Hole-board test; normally used for the assessment of measure anxiety, stress, neophilia, and emotionality in animals. The author mentioned as ‘The spontaneous locomotor activity was evaluated by using the hole-board test.’ Later mentioned as ‘….exploration of the holes (exploratory activity) by the mice.’. Need to rewrite with clear statements.
9. Section 4.10. Statistical analysis needs to describe clearly with what are the parameters tested with one-way ANOVA and what are the parameters tested with two-way ANOVA.
10. Reference section: The references are cited with old references. It needs to be improved with relevant recent references.
*****

Reviewer 2 Report
1. Sham controls are not performed in the experiments.
2. The volume of injection, 0.5 μl for each drug with a total of 1 μl injection within 15 minutes, is huge for the small size of the LC, as well as for the small size of the mouse brain. The injection drugs may spread to the other areas of the brainstem, even to the 4th ventricle. How large of the brain area do the drugs diffuse?
3. How is the concentration of drugs determined for the test?
4. What is DAPI? The description of the top panel, Fig. 2, is lacking. The immunofluorescence study shown in Fig. 2 was taken from the spinal cord, as indicated in the figure legend. On the other hand, the brainstem, but not the spinal cord, was used for H4R immunofluorescence, described in the Method. The whole section of either the brainstem or spinal cord should be displayed.
5. Hypothetical mechanisms underlying the effect of H4R antagonist on the role of H3R in pain modulation should be addressed.
6. Figure 5A. What is X-axis represent? Is the amount of food consumption cumulative or at different time periods?
7. The concentration of GSK189254 for LC injection is not indicated in the text.
8. Fig. 6A. What is the control? What is the “pretest”? Why did the number of falls decrease in the control 15 min after “pretest”? It is highly possible that a significant difference will be seen between “pretest” and after administration in both control and drug treatment. Please explain.
9. Reference #33 described the structure-activity relationships of H4R antagonists.
10. This study does not provide any evidence suggesting H3R and H4R localized on the opposite of the synaptic cleft, as the authors claimed (lines 241 and 242).
Reviewer 3 Report
Dear Authors,
Your work shows that histamine H4 receptors plays a role in the analgesic effect of GSK189254. The paper is quite well prepared. The results obtained are clearly presented in graphs and discussed in the text. The results show that the analgesic effect induced by the administration of GSK189254 was attenuated by the administration of the H4R antagonist (which is a selective H4R ligand (Ki = 26 nM) with a very weak affinity for H3R Ki = 14.1 μM).
The work is interesting and worthy of publication but needs some corrections.
IMPORTANT REMARKS
1. The paper at the end of the introduction lacks information about the compound GSK189254 itself. What is this compound, what is its affinity for histamine H3 receptors, is it a selective histamine H3 receptor antagonist (maybe it also has affinity for the histamine H4 receptor?) and why was it selected for this study (mention of the analgesic effect of GSK is only in the discussion) there is also no information that this compound has reached phase I of clinical trials (NCT00387413)
2. no explanation in the text as to why this GSK compound was tested at doses of 3 and 6 mg/kg p.o. and not, for example, 5 and 10 mg/kg p.o.?
3. lines 68 and 69 - in my opinion, the authors showed that blocking H4 receptors reduces the analgesic effect of GSK189254, and did not really show that H4R stimulation is selectively involved in the analgesic effect of GSK189254 - this sentence should be worded differently. Confirmation that H4R stimulation influences the analgesic effect of GSK would be a study in which an H4R agonist (e.g. VUF8430) and GSK were administered simultaneously. The authors showed only that HA and VUF8430 abolished mechanical and thermal allodynia, and that this effect was blocked by early administration of an H4R antagonist.
4. In the discussion there is no information that in the analgesic effect of H3 antagonists may also play a role their influence on sigma receptors (especially sigma 1). The effect of H3 antagonists on sigma 1 receptors was confirmed by Reddy at al (https://doi.org/10.1016/j.neuropharm.2018.10.028).
5. In the beginning of the discussion there is lack of information about JNJ10191584 (potent & selective H4 ligand)
6. In the text and in the drawings, JNJ10151984 is incorrectly written instead of JNJ10191584 e.g. line 90, 128, 140, 147, 157, 163, 165, 173, 180, 224, 250
minor comments
1. There should be an explanation of the abbreviations in the captions of figures 2-6, i.e. JNJ10191584 (JNJ), GSK189254 (GSK), VUF8430 (VUF)
2. line 83 is "bahviour" should be behaviour
3. line 117 - figure C and figure D – is GSK please write GSK6 it will be easier to analyse the drawing
4. line 126 is "GSK189254 3-6 mg/kg p.o.” what is incorrect as only 2 doses of GSK were tested (3 and 6 mg/kg o.p.) and not in this range; it should be GSK189254 (GSK) 3 and 6 mg.kg p.o.
5. line 186 – “at the highest effective dose” – add information that it was 6 mg/kg p.o.
6. line 206 – references 5 and 13 are not proper here as there is no information about the analgesic effect of GSK189254 in them; ref. 26 should be cited here
7. line 224 - reference 33 is inappropriate here. It does not support this sentence. Moreover it also does not describe this compound JNJ10191584. For the information about this compound should be cite Venable et al J Med Chem 2005, 48, 8289-98.
Round 2
Reviewer 2 Report
1. Sham controls are not performed in the experiments.
Sham controls, saline or 5%DMSO injection, should be performed in drug test studies.
2. The volume of injection, 0.5 μl for each drug with a total of 1 μl injection within 15 minutes, is huge for the small size of the LC, as well as for the small size of the mouse brain. The drug may also diffuse to the 4th ventricle. How large of the brain area do the drugs diffuse?
Please answer the comment, drug diffused to which areas of the brain (intra-LC injection).
3. How is the concentration of drugs determined for the study?
How is the concentration of GSK189254 determined for intra-LC injection?
4. What is DAPI? The description of the top panel, Fig. 2, is lacking. The immunofluorescence study shown in Fig. 2 was taken from the spinal cord, as indicated in the figure legend. On the other hand, the brainstem, but not the spinal cord, was used for H4R immunofluorescence, described in the Method. The whole section of either the brainstem or spinal cord should be displayed.
First, the description of the top panel, Fig. 2E, is not found in the text. Second, “Nuclei were stained with DAPI” is the only statement added in the figure legend. A more detailed description is needed. The low magnification of the spinal cord showing H4R immunofluorescence, in which high magnification of microphotographs are taken, should be displayed.
5. Hypothetical mechanisms underlying H4R antagonist reversed the effect of H3R on nociception should be addressed.
You said in your cover letter that the histamine released by GSK189254 could promote analgesia through activation of neuronal H4R in areas devoted to the modulation of pain perception (i.e spinal cord, LC).
In other words, application of GSK189254 increases histamine release, which activates H4R activity and subsequently causes coeruleospinal noradrenergic neuron activity resulting in suppressing pain. This statement should be added to “Discussion”. The Sigma 1 receptor does not involve in H3R-H4R on pain modulation. Delete it from “Discussion”.
6. Figure 5A. What is X-axis represent? Is the amount of food consumption cumulative or at different time periods?
“minutes” and “cumulative amount of food consumption” should be added to figure legend.
Round 3
Reviewer 2 Report
Symbols for the p value are missing in some figures.
Author Response
Symbols for the p values have been inserted in the figures.